# Acoustic indices as proxies for bird species richness in an urban green space in Metro Manila

**Skyla Dennise U. Diaz**[1]*, **Jelaine L. Gan**[1], **Giovanni A. Tapang**[2]

**1** Institute of Biology, College of Science, University of the Philippines Diliman, Quezon City, Philippines,
**2** National Institute of Physics, College of Science, University of the Philippines Diliman, Quezon City, Philippines

* sudiaz@up.edu.ph

## Abstract

We assessed eight acoustic indices as proxies for bird species richness in the National Science Complex (NSC), University of the Philippines Diliman. The acoustic indices were the normalized Acoustic Complexity Index (nACI), Acoustic Diversity Index (ADI), inverse Acoustic Evenness Index (1-AEI), Bioacoustic Index (BI), Acoustic Entropy Index (H), Temporal Entropy Index (Ht), Spectral Entropy Index (Hf), and Acoustic Richness Index (AR). Low-cost, automated sound recorders using a Raspberry Pi were placed in three sites at the NSC to continuously collect 5-min sound samples from July 2020 to January 2022. We selected 840 5-min sound samples, equivalent to 70 hours, through stratified sampling and pre-processed them before conducting acoustic index analysis on the raw and pre-processed data. We measured Spearman's correlation between each acoustic index and bird species richness obtained from manual spectrogram scanning and listening to recordings. We compared the correlation coefficients between the raw and pre-processed. *wav* files to assess the robustness of the indices using Fisher's z-transformation. Additionally, we used GLMMs to determine how acoustic indices predict bird species richness based on season and time of day. The Spearman's rank correlation and GLMM analysis showed significant, weak negative correlations between the nACI, 1-AEI, Ht, and AR with bird species richness. The weak correlations suggest that the performance of acoustic indices are dependent on various factors, such as the local noise conditions, bird species composition, season, and time of day. Thus, ground-truthing of the acoustic indices should be done before applying them in studies. Among the eight indices, the nACI was the best-performing index, performing consistently across sites and independently of season and time of day. We highlight the importance of pre-processing sound data from urban settings and other noisy environments before acoustic index analysis, as this strengthens the correlation between index values and bird species richness.

**Data Availability Statement:** All relevant data are within the manuscript and its Supporting Information files.

**Funding:** The author(s) received no specific funding for this work.

**Competing interests:** The authors have declared that no competing interests exist.

## Introduction

The presence of most bird species can be confirmed based on their distinct vocalizations [1]. Because of this, passive acoustic monitoring is suitable for studying bird communities [2, 3], and deploying sound recorders overcomes the logistical challenges posed by active point count and line-transect surveys [4]. If enough recorders are deployed, they can cover a greater area than human observers and can be left in the field for extended periods, especially if programmed to only record at specific time intervals [5]. In areas with mobile or Wi-Fi network connections, the collected digital sound files can automatically be transmitted to a central server for long-term data storage and analysis [6]. However, the challenge with acoustic surveys is the processing effort required to analyze extensive sound data collections and translate them into biodiversity measures that can give timely information relevant to species conservation and management [7].

Several ecoacoustic indices were proposed to summarize the acoustic features of soundscapes using the sound spectrum [8]. Moreover, acoustic indices are mathematical functions that measure the distribution of acoustic energy across time and frequency in a recording [9]. These indices enable the quick processing of sound data and measure the acoustic complexity, diversity, evenness, entropy, and richness of recordings. They are based on species diversity indices and are thought to characterize the variation in sound production in animal communities across time, serving as proxies for diversity metrics such as species abundance, richness, evenness, and diversity [10]. The ease with which these indices can be calculated using R packages, such as *soundecology* [11] and *seewave* [12], makes them attractive tools for rapid biodiversity assessment and monitoring.

Despite the growing usage of acoustic indices in ecoacoustic surveys, the relationships between index values and ground-truthed measures of biodiversity remain inconclusive [13]. The reference studies wherein these indices were developed reported positive correlations with bird species richness [14–18], but some studies found the opposite [19–21]. Retamosa Izaguirre et al. [22] and Moreno-Gómez et al. [23] evaluated acoustic indices as proxies for bird species richness, and both studies reported low to moderate correlations. Alcocer et al. [24] conducted a meta-analysis of acoustic indices as proxies for biodiversity and found that acoustic indices had a moderate positive relationship with the diversity metrics, such as species abundance, species richness, species diversity. As acoustic indices measure the amplitude and frequency properties of soundscapes, they cannot discern between species and may only weakly to moderately predict bird species richness.

Acoustic index values should be significantly correlated with the number of vocalizing bird species in recordings to reflect bird species richness accurately. However, field conditions can affect the performance of the acoustic indices [25, 26]. Noise and other environmental factors can confound the index values [27]. Outdoor environments have a diverse array of biophonies, such as sounds produced by birds, mammals, amphibians, and insects. Sounds from running water or rain falling through a canopy may also be present in the forest soundscape [28]. In rural and urban areas, geophony from breezes and drizzles are sources of bias when using acoustic indices [20]. At sites near farmlands, roads, or human settlements, sounds generated by vehicles, sirens, machines, and human speech may affect acoustic index values [27]. Unavoidable ambient sounds highlight the importance of pre-processing sound data [17, 18] and ground-truthing of the acoustic indices (i.e., validation of index values with survey data) before applying them in studies.

We assessed eight acoustic indices as proxies for bird species richness in an urban green space in Metro Manila. The anthropause [29], or slowdown of human activity during the COVID-19 pandemic, was an opportune time to measure bird vocalizations with reduced

human disturbance. The acoustic indices assessed in the study were the normalized Acoustic Complexity Index (nACI), Acoustic Diversity Index (ADI), inverse Acoustic Evenness Index (1-AEI), Bioacoustic Index (BI), Acoustic Entropy Index (H), Temporal Entropy Index (Ht), Spectral Entropy Index (Hf), and Acoustic Richness Index (AR). We measured the correlation between each acoustic index and the bird species richness obtained from manual spectrogram scanning and listening to recordings. The correlations using two datasets, the unprocessed (raw) and the pre-processed sound files, were also compared to assess the robustness of the acoustic indices. We used generalized linear mixed models (GLMMs) to examine how acoustic indices predict bird species richness in pre-processed 5-min sound samples dependently on season and time of day. We hypothesized that if acoustic index values are significantly correlated with the number of species heard, then the selected indices can be used as proxies for bird species richness in the field.

## Materials and methods

We conducted the study at the National Science Complex (NSC) in the University of the Philippines Diliman (UPD) campus, one of the last urban green spaces in Metro Manila, Philippines. The study area is home to 42 bird species, including endemic, resident, and migrant bird species [30]. Located within the NSC is the one-hectare UP Biology—EDC BINHI Threatened Species Arboretum, home to endemic and threatened tree species [31, 32] and various bird species [33]. The anthropause [29], or slowdown of human activity during the COVID-19 pandemic, provided an opportune time to measure bird vocalizations with reduced human disturbance. With COVID-19 restrictions and suspended face-to-face classes, entering the campus was restricted to authorized personnel only. Metro Manila has a Type I climate, marked by two pronounced seasons: the dry season from November to April, and the wet season from May to October. The months of June to September receive the highest rainfall [34].

### Ethics statement

No specific ethical approval was needed because automated sound recorders were used to collect data remotely.

### Study area

We established three recording sites, namely the Institute of Biology (A), the College of Science Administrative Building (B), and the College of Science Library (C) (Fig 1). The recording sites were selected in consideration of Wi-Fi and electricity access. The sites are adequately spaced (>100m) from each other to avoid overlapping between recording points. Site A directly faces the UP Biology—EDC BINHI Threatened Species Arboretum, but it is disturbed by ongoing construction and maintenance work. Site B is approximately 50 meters from the closest tree line and is surrounded by a road network. Site C is enclosed by vegetation, and the recorder is less than 10 meters from the closest tree line and about 30 meters from the nearest road. Guard posts are also located at the entrance going to Site C.

### Acoustic data collection

At each recording point, we placed a low-cost sound recorder built using an omnidirectional USB microphone attached to a Raspberry Pi 3 Model B+ (Fig 1) [37]. All three systems, running with Raspberry Pi OS, were programmed to record 5-min sound clips every 6-min, using a sampling rate of 44.1 kHz. This resulted in a 1-min gap between recording bouts. The systems were recording continuously from July 2020 to January 2022 to cover the wet and dry

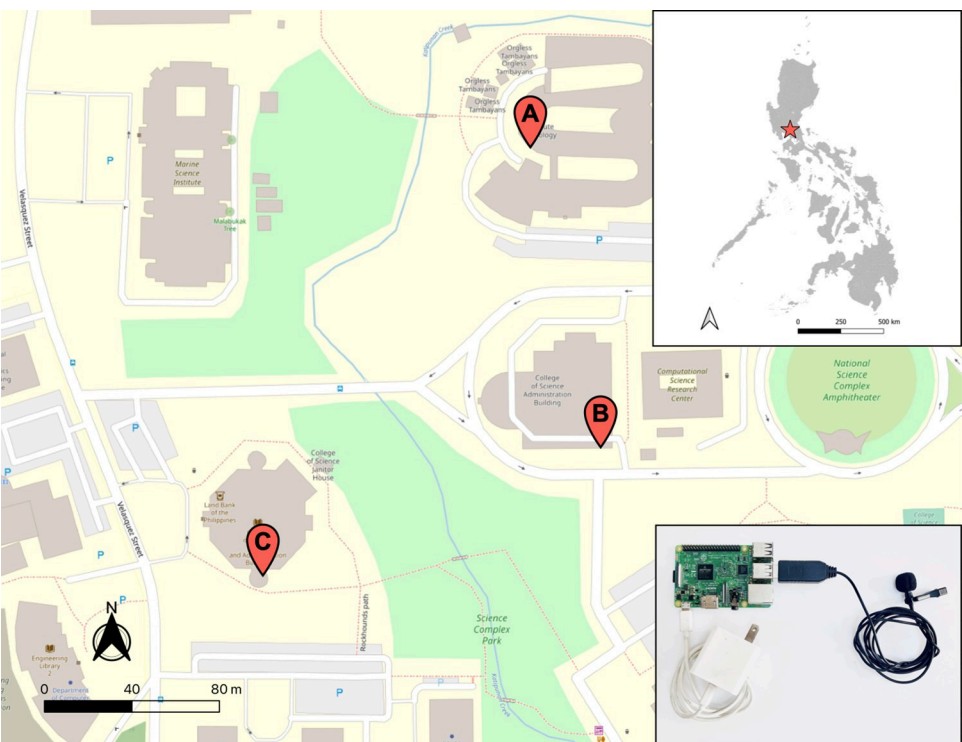

**Fig 1. Map of three sound recording sites within the National Science Complex, University of the Philippines Diliman, and its relative location in the Philippines.** This map was generated with data from OpenStreetMap [35] and Wikimedia Commons [36]. (A) Institute of Biology, (B) College of Science Administrative Building, and (C) College of Science Library. The inset shows the recording system (USB microphone attached to Raspberry Pi 3 Model B+).

seasons and account for migratory birds. These were done using the program *arecord* [38], called in a script that also uploads the WAV files to Google Drive for long-term data storage and analysis.

Low-cost recorders may have varying sensitivities, resulting in different responses in terms of frequency range and signal intensity [39]. Thus, we tested the response of each low-cost recorder across a range of frequencies. Using version 3.0.0 of Audacity® recording and editing software [40], we generated a 30-sec tone from 1 Hz to 10,000 Hz and played it through a loudspeaker held at breast level, gradually increasing the distance from the recorder. We plotted the spectrum data of the 30-sec generated tone against the average response of each recorder (S1 Fig). The frequency response was not flat over the entire frequency range, and higher frequencies were increasingly attenuated, except for recorder A (S1 Fig).

## Data pre-processing

We randomly selected 840 *.wav* files from the database based on the YYYY-MM-DD-HH-mm timestamp, totaling 4,200 minutes (≈70 hours). Through stratified sampling, we selected 20 sound samples from each hour between 5:00 AM and 6:00 PM to cover the daily activity period, including the dawn and dusk chorus. This resulted in a total of 280 samples for each recording point. We did not remove recordings with heavy rain or strong wind from the sample.

We performed pre-processing on the 840 *.wav* files by adding a high-pass filter (400.0 Hz) and performed subsequent noise reduction and signal amplification in Audacity® macros [40] to reduce anthropogenic and geophonic background noise [18]. We used the default

settings to streamline the processing of multiple.*wav* files, and upon visual inspection, the noise reduction setting (12 dB) was enough to reduce the unwanted background noise.

## Manual spectrogram scanning and listening to recordings

We used the 840 pre-processed sound samples for manual spectrogram scanning and listening to recordings. We viewed each 5-min sound sample in Audacity®, using Spectrogram View set to Linear Scale, with the Minimum Frequency set to 0 kHz and the Max Frequency set to 8 kHz (Fig 2). S. Diaz was the primary observer who analyzed all the recordings to reduce variability between observers. We listened to the selected recordings and identified all bird vocalizations up to the species level; thus, each WAV file had a list of species heard. Sound files from online repositories of recorded avian vocalizations, such as Xeno-canto [41] and eBird [42], were used as references in identifying bird species in the recordings. In addition, expert consultation from experienced birders (J. Gan, C. Española, and A. Constantino) helped in further species identification. Unidentified vocalizations with distinct time-frequency characteristics were still included as sonospecies.

The limitation of our study is that we did not measure bird species abundance because of the difficulty of telling apart individuals based simply on their calls. Hence, we focused on the relationship of acoustic indices with bird species richness.

## Acoustic index analysis

We used two data sets for the acoustic index analysis: raw (direct from recording) and pre-processed.*wav* files. We calculated eight acoustic indices: Acoustic Complexity Index (ACI) [14], Acoustic Diversity Index (ADI) [15], Acoustic Evenness Index (AEI) [15], Bioacoustic Index (BI) [16], Acoustic Entropy Index (H) [17], Temporal Entropy Index (Ht) [17], Spectral Entropy Index (Hf) [17], and Acoustic Richness Index (AR) [18] using the *soundecology* [11] and *seewave* [12] R packages [43]. We used a sampling frequency of 44,100 Hz and a fast Fourier transform (FFT) window of 512 points (S1 Table). This corresponds to a frequency resolution of FR = 86.133 Hz and a time resolution of TR = 0.0116 s.

All acoustic indices, except H and AR, were calculated using the *multiple_sounds* function in *soundecology*. We used the default settings for the ACI, ADI, AEI, and BI (S1 Table). The

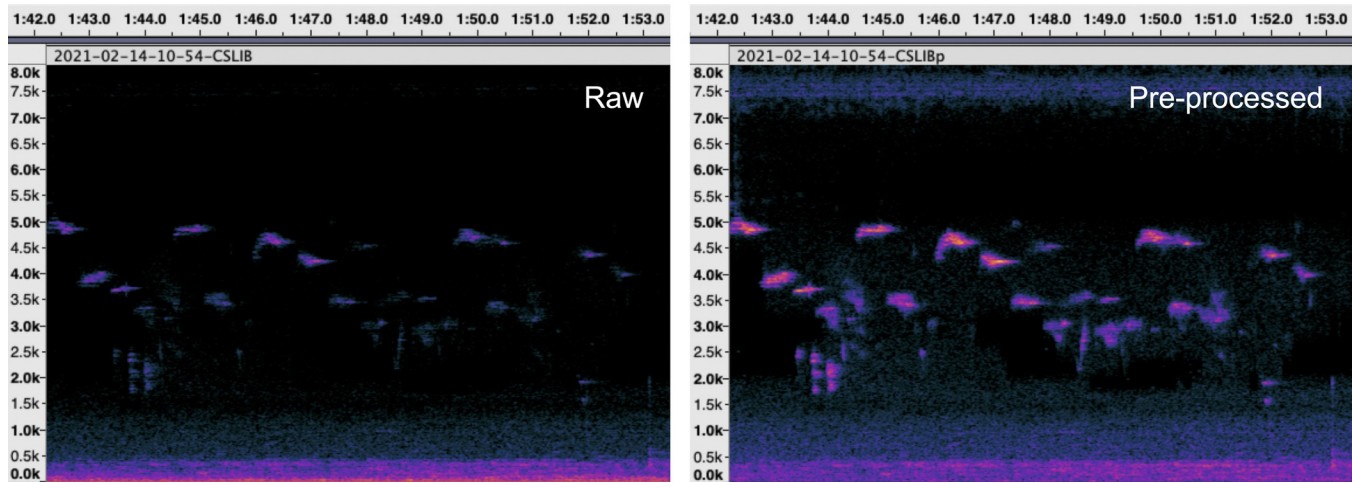

**Fig 2. Sample spectrograms for raw and pre-processed.*wav* files opened in Audacity® Spectrogram View.** The x-axis represents the time expressed in seconds, while the y-axis represents the frequency expressed in kHz. Spectrogram parameters: linear scale, 8 kHz maximum frequency, and window size = 4096. Amplitude ranges from -60 dB to -44 dB (raw) and -60 dB to -29 dB (pre-processed).

maximum frequency of ADI and AEI was set to 10 kHz, and the frequency step was set to 1 kHz. The minimum frequency of BI was set to 2 kHz, and the maximum frequency was set to 8 kHz. We calculated the H and AR in *seewave* using the default settings (S1 Table). We extracted the Ht from the results of AR and each H value by its corresponding Ht value to obtain Hf.

Since the results given by ACI are accumulative (i.e., very long samples will return very large values) [11], we divided each ACI value by the ACI value of a silent recording (= ACI/ 8843.141) to get a range of values easier to compare, herein referred to as normalized Acoustic Complexity Index (nACI). We also calculated the inverse Acoustic Evenness Index (1-AEI) to facilitate comparability with the ADI [21]. Other indices had no further calculations.

**Normalized Acoustic Complexity Index (nACI).** The ACI is calculated from a spectrogram divided into frequency bins and temporal steps [14]. This index calculates the absolute difference between adjacent sound intensity values in each frequency bin. The absolute differences for all frequency bins are summed to obtain the ACI for the entire recording [14]. Based on this formula, high ACI values correspond to intensity modulations with the assumption that bird vocalizations dominated the recording. In contrast, low ACI values correspond to constant intensities, increased insect buzz, and anthrophony (i.e., human-generated sounds). The ACI correlated positively with the number of bird vocalizations in a beech forest in Italy [14].

**Acoustic Diversity Index (ADI) and Inverse Acoustic Evenness Index (1-AEI).** The ADI and AEI are calculated on a matrix of amplitude values extrapolated from a spectrogram divided into frequency bins. Both indices first obtain the proportion of amplitude values in each bin above a dB threshold [15]. The ADI applies the Shannon diversity index to these values, whereas the AEI applies the Gini index. A higher Shannon's index indicates higher diversity, whereas a higher Gini index indicates lower evenness and diversity. Consequently, ADI values should correlate positively and AEI values negatively with bird species richness. In Indiana, USA, the ADI values were higher, and AEI values were lower during the dawn and dusk choruses, indicating the presence of more vocalizing bird species during these periods [15]. We calculated the inverse of AEI, herein referred to as the inverse Acoustic Evenness Index (1-AEI) to facilitate comparability between the two indices [21]. Higher ADI and 1-AEI should indicate higher frequency occupation of the spectrum and higher sound diversity.

**Bioacoustic Index (BI).** The BI is calculated based on a spectrogram generated from a fast Fourier transform (FFT). This index obtains the area of the FFT (i.e., total sound energy) between 2 and 8 kHz, which is the typical frequency range of bird vocalizations [16]. The BI was strongly and positively correlated with avian abundance in Hawaii [16].

**Acoustic Entropy Index (H): Temporal Entropy Index (Ht) and Spectral Entropy Index (Hf).** The H is calculated as the product of Ht and Hf, where Ht represents the Shannon entropy of the amplitude envelope, and Hf represents the Shannon entropy of the frequency spectrum [17]. A flatter amplitude envelope, or low entropy, indicates constant intensities, whereas higher H values indicate more intensity modulations [17]. In Tanzanian lowland coastal forests, H increased with bird species richness [17].

**Acoustic Richness Index (AR).** The AR is based on Ht and the median of the amplitude envelope (M) [18]. Lower AR values (i.e., flatter amplitude envelope and lower entropy) indicate lower acoustic richness and the presence of fewer bird species. In contrast, higher AR values correspond to increased amplitude modulations, higher acoustic richness, and more bird species. In France, the AR correlated positively with bird species richness [18].

## Statistical analysis

**Spearman's correlation.** We calculated Spearman's rank correlation between each acoustic index and the bird species richness obtained from manual spectrogram scanning and

listening to recordings. We used two data sets: 840 raw and 840 pre-processed.*wav* files, herein referred to as pooled raw data and pooled pre-processed data. Additionally, we calculated Spearman's rank correlation separately for each recording point. We calculated Fisher's z-transformation to compare the correlations between the raw and pre-processed datasets.

**Generalized linear mixed models (GLMMs).** We used generalized linear mixed models (GLMMs) to examine how acoustic indices predict bird species richness in pre-processed 5-min sound samples depending on season and time of day. For each sound sample, we specified the acoustic index as the dependent variable, bird species richness, season, and time of day as fixed effects, and the recording point as the random effect. We fitted a set of three models for each acoustic index, which included the full model and fixed effects models in R using the *lme4* [44] and *glmmTMB* [45] packages (S2 Table). We used GLMMs to fit nACI, ADI, and BI with Gaussian distribution and identity link function [46]; AR with Gaussian distribution and log link function; and 1-AEI, H, Ht, and Hf with beta distribution [47]. We computed the corrected Akaike Information Criterion ($AIC_c$) using the *AICcmodavg* [48] R package and chose the model with the lowest $AIC_c$ value as the best model for each acoustic index [49]. We considered all models with $\Delta AIC_c < 4$ as equally plausible [49]. Lastly, we compared the plausible models using likelihood-ratio tests (LRT).

## Results

### Summary of bird species heard

From the 840 pre-processed 5-min sound samples analyzed, we heard 21 bird species from 20 families in the recordings from the NSC (Table 1). This represented only 52.5% of the 40

**Table 1. Summary of bird species heard and number of sound samples each species was detected out of 280 samples per recording point.**

| Family | Common Name | Scientific Name | Site A | Site B | Site C |
|---|---|---|---|---|---|
| Acanthizidae | Golden-bellied Gerygone | *Gerygone sulphurea* | 32 | 127 | 143 |
| Alcedinidae | Collared Kingfisher | *Todiramphus chloris* | 46 | 76 | 105 |
| Ardeidae | Black-crowned Night Heron | *Nycticorax nycticorax* | 2 | 0 | 2 |
| Artamidae | White-breasted Woodswallow | *Artamus leucorynchus* | 3 | 2 | 0 |
| Campephagidae | Pied Triller | *Lalage nigra* | 2 | 5 | 3 |
| Caprimulgidae | Philippine Nightjar* | *Caprimulgus manillensis* | 2 | 1 | 2 |
| Columbidae | Zebra Dove | *Geopelia striata* | 1 | 2 | 11 |
| Corvidae | Large-billed Crow | *Corvus macrorhynchos* | 32 | 127 | 143 |
| Dicaeidae | Red-keeled Flowerpecker* | *Dicaeum australe* | 24 | 4 | 35 |
|  | Pygmy Flowerpecker* | *Dicaeum pygmaeum* | 0 | 0 | 1 |
| Laniidae | Brown Shrike | *Lanius cristatus* | 13 | 6 | 1 |
| Megalaimidae | Coppersmith Barbet | *Psilopogon haemacephalus* | 0 | 7 | 31 |
| Muscicapidae | Philippine Magpie-robin* | *Copsychus mindanensis* | 5 | 1 | 1 |
| Nectariniidae | Olive-backed Sunbird | *Cinnyris jugularis* | 69 | 47 | 28 |
| Oriolidae | Black-naped Oriole | *Oriolus chinensis* | 130 | 121 | 86 |
| Passeridae | Eurasian Tree Sparrow | *Passer montanus* | 84 | 49 | 86 |
| Picidae | Philippine Pygmy Woodpecker* | *Yungipicus maculatus* | 15 | 3 | 4 |
| Psittaculidae | Philippine Hanging Parrot* | *Loriculus philippensis* | 21 | 16 | 33 |
| Pycnonotidae | Yellow-vented Bulbul | *Pycnonotus goiavier* | 115 | 155 | 97 |
| Rhipiduridae | Philippine Pied Fantail* | *Rhipidura nigritorquis* | 36 | 21 | 23 |
| Zosteropidae | Lowland White-eye | *Zosterops meyeni* | 18 | 66 | 8 |

From July 2020 to January 2022, 21 species from 20 families were heard in the National Science Complex. Asterisk (*) indicates Philippine endemic species.

known species sighted at the NSC based on recent eBird checklists [50] submitted from July 2020 to January 2022. However, two species, namely the Philippine Nightjar and Pygmy Flowerpecker, were heard in the recordings but not included in the list because there is insufficient eBird data available for both species. Of the 21 detected bird species, seven (33%) are Philippine endemics (Table 1). Thirteen are resident species. The Brown Shrike is the only migrant species. Seventeen (81.0%) bird species were found at all three sites. Given the proximity of the three sites, the bird community across them is most likely shared. We assume that the bird abundance and bird richness are similar across the three sites.

An uneven distribution of species was heard in the recordings (Fig 3). The highest number of bird species heard in a single recording was seven. However, this only comprised twelve (1.43%) of the 840 pre-processed *.wav* files. Seventy percent (588 *.wav* files) had one to four species recorded, and 15% (129 *.wav* files) had zero bird species heard.

In 840 pre-processed 5-min sound samples, the highest mean richness was observed from recordings between 6:00 AM and 8:00 AM, indicating the dawn chorus (Fig 4). The dusk chorus was less pronounced.

In addition to bird sounds, we also detected various other biotic sounds, including those produced by mammals, amphibians, reptiles, and insects. These include the sounds of chickens, dogs, geckos, frogs, sheep, and crickets. We also identified anthroponies, such as human speech, construction noise, and road traffic, as well as geophonies, such as rain and wind. Although we detected other non-avian sounds in our recordings, we did not take their frequencies into account.

## Spearman's correlation between acoustic indices and bird species richness

The Spearman's correlation and Fisher's z-transformation analyses revealed that bird species richness correlated more strongly with the pre-processed data than with the raw data (Tables 2

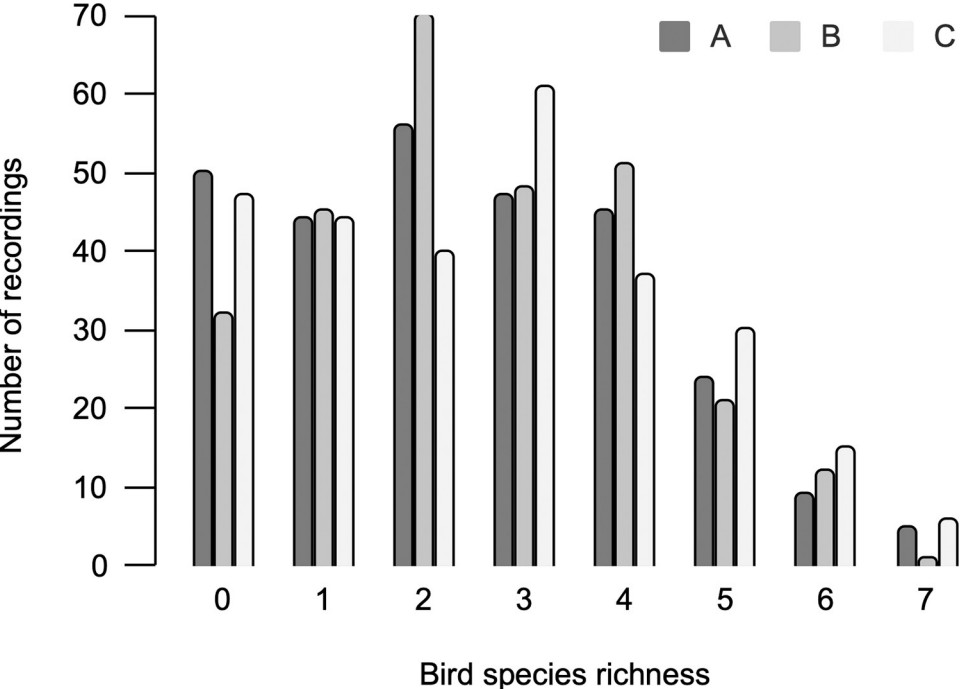

**Fig 3. Bird species richness and number of recordings per site in 840 pre-processed 5-min sound samples.**

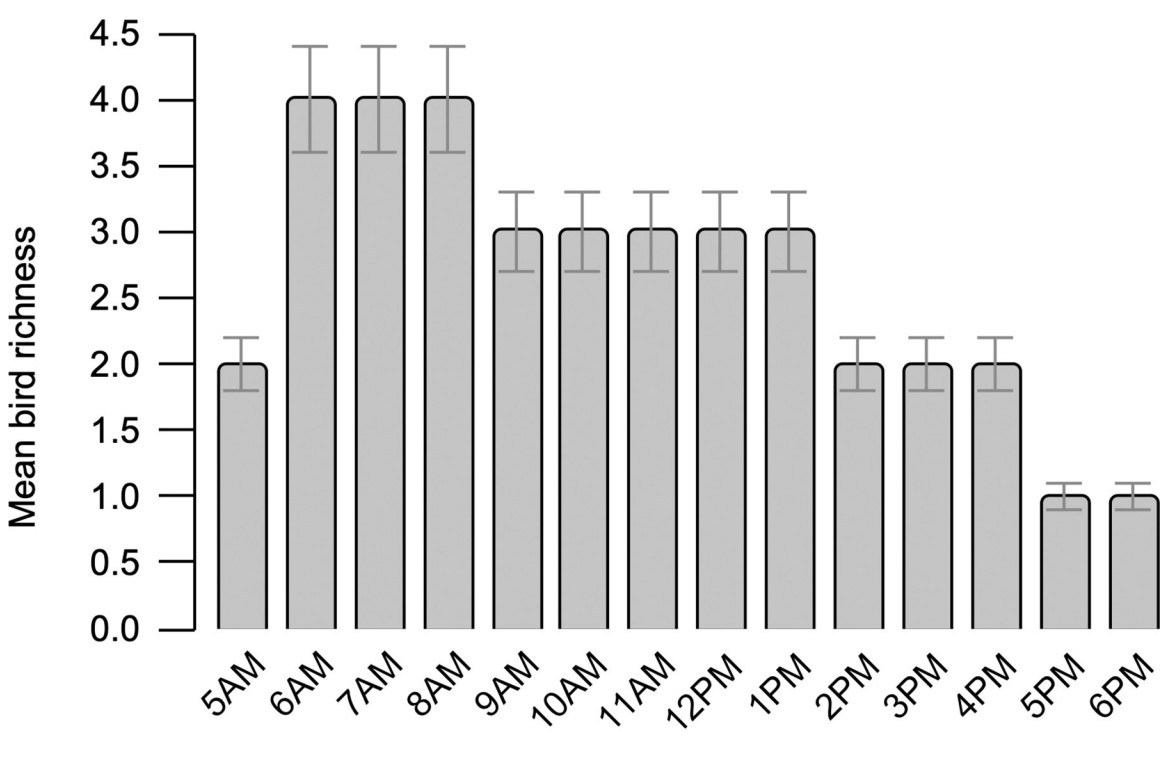

**Fig 4. Mean bird species richness across time from 5:00 AM to 6:00 PM in 840 pre-processed 5-min sound samples.**

and 3). Regarding the pooled pre-processed data, five acoustic indices, namely the nACI, ADI, 1-AEI, Ht, and AR, had significant but weak negative correlations with the number of bird species heard in the recordings (Table 2). The three other acoustic indices, BI, H, and Hf, had no significant correlation with bird species richness.

**Table 2. The Spearman's rank correlation coefficients between acoustic indices and bird species richness.**

| Site | nACI | ADI | 1-AEI | BI | H | Ht | Hf | AR |
|------|------|-----|-------|-----|---|-----|-----|-----|
| | | | | *Raw (unprocessed)* | | | | |
| Pooled | **-0.10**\*\* | -0.02 | -0.02 | -0.01 | **-0.07**\* | 0.02 | **-0.07**\* | -0.001 |
| A | **-0.25**\*\*\* | **-0.14**\* | **-0.15**\* | -0.01 | **-0.19**\*\* | 0.02 | **-0.19**\*\* | -0.09 |
| B | -0.05 | -0.01 | -0.01 | **-0.16**\*\* | -0.01 | 0.08 | -0.02 | 0.01 |
| C | -0.04 | **0.15**\* | **0.17**\*\* | 0.05 | 0.01 | 0.12 | 0.004 | 0.09 |
| | | | | *Pre-processed* | | | | |
| Pooled | **-0.15**\*\*\*\* | **-0.09**\*\* | **-0.09**\*\* | -0.01 | 0.02 | **-0.10**\*\* | 0.05 | **-0.11**\*\* |
| A | **-0.25**\*\*\*\* | **-0.20**\*\* | **-0.20**\*\*\* | 0.01 | -0.08 | -0.05 | -0.05 | **-0.13**\* |
| B | -0.05 | -0.10 | -0.10 | **-0.18**\*\* | **0.15**\* | -0.03 | **0.16**\*\* | -0.04 |
| C | **-0.20**\*\*\* | -0.05 | -0.05 | 0.06 | 0.05 | **-0.20**\*\*\* | 0.10 | **-0.16**\*\* |

Based on the pooled pre-processed data, the nACI, ADI, 1-AEI, Ht, and AR had significant but weak negative correlations with bird species richness. Asterisks and bold indicate significance levels

\* $p < 0.05$

\*\* $p < 0.01$

\*\*\* $p < 0.001$.

**Table 3. The results of Fisher's z-transformation comparing Spearman's rank correlation coefficients for the raw and pre-processed data sets.**

| Index | $z_{raw}$ | $z_{pre-processed}$ | $z_{diff}$ | t | p |
|---|---|---|---|---|---|
| nACI | -0.104 | -0.148 | -0.044 | -1.817 | 0.069 |
| ADI | -0.023 | -0.092 | -0.069 | -2.839 | <0.01 |
| 1-AEI | -0.015 | -0.093 | -0.077 | -3.165 | <0.01 |
| BI | -0.007 | -0.006 | 0.001 | 0.040 | 0.968 |
| H | -0.072 | 0.024 | 0.096 | 3.927 | <0.001 |
| Ht | 0.022 | -0.099 | -0.121 | -4.942 | <0.001 |
| Hf | -0.074 | 0.052 | 0.125 | 5.133 | <0.001 |
| AR | -0.001 | -0.111 | -0.110 | -4.514 | <0.001 |

Pre-processing significantly strengthened the ADI, 1-AEI, Ht, and AR correlations.

Fisher's z-transformation results showed no significant improvement in the correlation between the nACI and bird species richness (Table 3). Pre-processing significantly strengthened the correlations for ADI, 1-AEI, Ht, and AR, whereas the correlations for H and Hf became significantly weaker (Table 3).

### Generalized linear mixed models (GLMMs)

The 1-AEI, Ht, and AR predicted bird species richness in pre-processed 5-min sound samples, and they showed significant differences with season and time of day (Table 4). Specifically, the 1-AEI, Ht, and AR were significantly higher during the wet season than the dry season. The nACI also predicted bird species richness; however, we found no significant differences associated with season and time of day (Table 4). Conversely, the ADI, BI, H, and Hf did not significantly estimate bird species richness (Table 4). We found that season and time of day were stronger predictors for these indices.

## Discussion

### Acoustic indices and bird species richness

Our study showed that the nACI, 1-AEI, Ht, and AR estimated bird species richness in pre-processed 5-min sound samples, showing a significant negative relationship between index values and bird richness. We found nACI to be the most robust, as it predicted bird species richness independently of season and time of day and consistently across sites. The negative relationship between nACI and bird species richness may be attributed to the overlapping vocalizations of multiple birds in the recordings. This leads to reduced differences in sound intensities and lower nACI values. The ACI also showed a significant negative correlation with bird species richness in a grassland ecosystem in the U.S.A [51] and a tropical rainforest in Costa Rica [52]. The negative correlation was explained by less complex bird calls at the study sites and the presence of dominant species, which resulted in less variable sound intensities [51, 52].

While the nACI was a more robust index, the 1-AEI, Ht, and AR were reliable predictors of bird species richness in urban settings when pre-processing was applied to the *wav* files. After pre-processing, the 1-AEI, Ht, and AR showed stronger correlations with bird species richness (Table 3). Therefore, using the 1-AEI, Ht, and AR with pre-processed data can lead to better predictions of bird species richness in green spaces and other urban environments.

The negative correlation between 1-AEI and bird species richness could be due to bird species occupying a wide frequency range, resulting in high index values. Since recordings with

**Table 4. The results of GLMM examining how eight acoustic indices reflect bird species richness, season, and time of day, using the 840 pre-processed 5-min sound samples dataset.**

|  | Coefficient | SE | t | p |
|---|---|---|---|---|
| *Normalized Acoustic Complexity Index (nACI)* | | | | |
| Intercept | 1.039 | 0.011 | 94.470 | <0.001 |
| Richness | -0.003 | 0.001 | -4.698 | <0.001 |
| Season [wet] | -0.003 | 0.002 | -1.422 | 0.156 |
| Hour | -0.0004 | 0.0002 | -1.651 | 0.099 |
| *Acoustic Diversity Index (ADI)* | | | | |
| Intercept | 1.787 | 0.233 | 7.672 | <0.01 |
| Richness | -0.013 | 0.010 | -1.348 | 0.178 |
| Season [wet] | 0.253 | 0.033 | 7.652 | <0.001 |
| Hour | -0.014 | 0.004 | -3.285 | <0.01 |
| *Inverse Acoustic Evenness Index (1-AEI)* | | | | |
| Intercept | 0.574 | 0.447 | 1.283 | 0.200 |
| Richness | -0.068 | 0.020 | -3.417 | <0.001 |
| Season [wet] | 0.369 | 0.067 | 5.529 | <0.001 |
| Hour | -0.021 | 0.009 | -2.419 | <0.05 |
| *Bioacoustic Index (BI)* | | | | |
| Intercept | 7.735 | 2.552 | 3.031 | 0.087 |
| Richness | -0.086 | 0.065 | -1.335 | 0.182 |
| Season [wet] | 1.065 | 0.214 | 4.966 | <0.001 |
| Hour | -0.079 | 0.028 | -2.822 | <0.01 |
| *Acoustic Entropy Index (H)* | | | | |
| Intercept | 1.714 | 0.152 | 11.282 | <0.001 |
| Richness | 0.001 | 0.008 | 0.072 | 0.942 |
| Season [wet] | 0.075 | 0.028 | 2.693 | <0.01 |
| Hour | -0.008 | 0.004 | -2.230 | <0.05 |
| *Temporal Entropy Index (Ht)* | | | | |
| Intercept | 3.915 | 0.162 | 24.201 | <0.001 |
| Richness | -0.030 | 0.010 | -2.991 | <0.01 |
| Season [wet] | 0.125 | 0.034 | 3.667 | <0.001 |
| Hour | -0.009 | 0.004 | -2.103 | <0.05 |
| *Spectral Entropy Index (Hf)* | | | | |
| Intercept | 1.854 | 0.150 | 12.340 | <0.001 |
| Richness | 0.006 | 0.009 | 0.684 | 0.494 |
| Season [wet] | 0.064 | 0.029 | 2.197 | <0.05 |
| Hour | -0.008 | 0.004 | -2.137 | <0.05 |
| *Acoustic Richness Index (AR)* | | | | |
| Intercept | -1.066 | 0.135 | -7.880 | <0.001 |
| Richness | -0.072 | 0.018 | -4.021 | <0.001 |
| Season [wet] | 0.580 | 0.068 | 8.588 | <0.001 |
| Hour | -0.038 | 0.008 | -5.028 | <0.001 |

We specified the acoustic index as the dependent variable, bird species richness, season, and time of day as fixed effects, and the recording point as the random effect.

heavy rain and intense anthrophony were also not excluded, these recordings yielded high 1-AEI values, despite having little to no audible bird species.

The Ht also had a significant negative relationship with bird species richness in a rainforest in Chile [23]. The low diversity of birds in the recordings and the presence of one or few

dominant species may explain the negative correlation of the Ht with bird species richness. If only a few bird species vocalize at a time, the amplitude modulations become more distinct, and the envelope becomes less flat, returning higher Ht values.

In contrast, the BI, H, and Hf showed inconsistent and poor performance in predicting bird richness. The correlations remained inconsistent across three sites, even after pre-processing. Therefore, we do not recommend using the BI, H, and Hf as proxies for bird species richness in urban settings.

Given the performance of the eight acoustic indices assessed, we recommend using only the nACI, 1-AEI, Ht, and AR as proxies for bird species richness in urban settings. However, ground-truthing of the acoustic indices should be conducted before applying them to studies in different environments as their performance depends on various factors including, the local noise conditions, bird species composition of the site, season, and time of day.

Although most bird species in the NSC were heard at all three sites, different local noise conditions due to the construction and maintenance work at site A, the proximity of site B to the road, or site C to the tree line, resulted in inconsistent correlations between index values and bird species richness. However, pre-processing strengthened the correlation between select acoustic indices (ADI, 1-AEI, Ht, AR) and bird richness. Our study highlights the importance of pre-processing sound data from urban and noisy environments before using acoustic indices. If background noise has been preliminarily removed from recordings by applying amplitude threshold cut-off filters (e.g., high-pass filter), noise reduction, and signal amplification, then acoustic indices can perform consistently across diverse acoustic conditions and accurately reflect bird species richness [17, 18].

## Conclusions

We demonstrate passive acoustic monitoring using low-cost sound recorders to study urban bird communities in the Philippines and attempt to address the challenge of processing and analyzing extensive sound data collections. To our knowledge, this is the first study assessing acoustic indices as proxies for bird species richness in an urban green space in the country. Among the eight acoustic indices assessed in the study, the nACI was the best and most robust, while the 1-AEI, Ht, and AR were also good predictors of bird species richness. We emphasize the importance of ground-truthing the indices before applying them to studies. Research exploring the acoustic indices and their relation to bird species richness is needed to understand the inconsistent correlations fully.

As the frequency response of our recorders was not flat over the entire frequency range, which impacts the calculation of acoustic indices, low-cost recorders should be calibrated before use in ecoacoustic studies to offset this limitation. We also recognize the importance of pre-processing sound data, such as applying high-pass filters, noise reduction, and signal amplification, to account for differences among recorders, remove unwanted ambient sounds, and improve the detection of bird species.

As acoustic indices were created to measure different characteristics of soundscapes quickly, they are not species-specific and lack sound recognition capabilities. For this reason, machine learning via convolutional neural networks (CNN) is increasingly being used in bioacoustic studies for birdsong classification [53–55]. Neural networks trained on Xeno-canto and Macaulay Library recordings have successfully classified North American and European bird species [53]. This remains to be explored in Philippine bird species. The use of other analysis, such as principal component analysis (PCA) [56] and cluster analysis [57], also help discriminate among different soundscapes in urban areas based on ecoacoustic indices.

## Supporting information

**S1 Fig. The frequency response of recorders A-C with respect to the generated tone (black line) in Audacity® [40].**
(PDF)

**S1 Table. Summary of parameters used in the calculation of acoustic indices.**
(PDF)

**S2 Table. The results of GLMM examining how eight acoustic indices reflect bird species richness, season, and time of day, using the dataset of 840 pre-processed 5-min sound samples.**
(PDF)

**S3 Table. The results of the acoustic index analysis for 840 raw 5-min sound samples.**
(PDF)

**S4 Table. The results of the acoustic index analysis for 840 pre-processed 5-min sound samples.**
(PDF)

## Acknowledgments

The authors appreciate the help of expert birders, Dr. Carmela P. Española and Adrian Constantino, in species identification.

## Author Contributions

**Conceptualization:** Skyla Dennise U. Diaz, Jelaine L. Gan, Giovanni A. Tapang.

**Data curation:** Skyla Dennise U. Diaz, Jelaine L. Gan, Giovanni A. Tapang.

**Formal analysis:** Jelaine L. Gan, Giovanni A. Tapang.

**Investigation:** Skyla Dennise U. Diaz, Jelaine L. Gan, Giovanni A. Tapang.

**Methodology:** Skyla Dennise U. Diaz, Jelaine L. Gan, Giovanni A. Tapang.

**Project administration:** Jelaine L. Gan, Giovanni A. Tapang.

**Resources:** Giovanni A. Tapang.

**Software:** Giovanni A. Tapang.

**Supervision:** Jelaine L. Gan, Giovanni A. Tapang.

**Validation:** Skyla Dennise U. Diaz, Jelaine L. Gan, Giovanni A. Tapang.

**Visualization:** Skyla Dennise U. Diaz, Jelaine L. Gan, Giovanni A. Tapang.

**Writing – original draft:** Skyla Dennise U. Diaz.

**Writing – review & editing:** Skyla Dennise U. Diaz, Jelaine L. Gan, Giovanni A. Tapang.

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
