## [Decision Letter · Decision Letter 0]

2 Feb 2023

PONE-D-22-34645Acoustic indices as proxies for bird species richness in an urban green space in Metro ManilaPLOS ONE

Dear Dr. Diaz,

Thank you for submitting your manuscript to PLOS ONE. After careful consideration, we feel that it has merit but does not fully meet PLOS ONE’s publication criteria as it currently stands. Therefore, we invite you to submit a revised version of the manuscript that addresses the points raised during the review process.

We look forward to receiving your revised manuscript.

Kind regards,

Cihun-Siyong Alex Gong, Ph.D.

Academic Editor

PLOS ONE

Journal Requirements:

2. We note that Figure 1 in your submission contain [map/satellite] images which may be copyrighted. All PLOS content is published under the Creative Commons Attribution License (CC BY 4.0), which means that the manuscript, images, and Supporting Information files will be freely available online, and any third party is permitted to access, download, copy, distribute, and use these materials in any way, even commercially, with proper attribution. For these reasons, we cannot publish previously copyrighted maps or satellite images created using proprietary data, such as Google software (Google Maps, Street View, and Earth). For more information, see our copyright guidelines: http://journals.plos.org/plosone/s/licenses-and-copyright.

Reviewers' comments:

Reviewer's Responses to Questions

**Comments to the Author**

1. Is the manuscript technically sound, and do the data support the conclusions?

Reviewer #1: Yes

Reviewer #2: Partly

Reviewer #3: Partly

2. Has the statistical analysis been performed appropriately and rigorously? 

Reviewer #1: Yes

Reviewer #2: No

Reviewer #3: No

3. Have the authors made all data underlying the findings in their manuscript fully available?

Reviewer #1: Yes

Reviewer #2: No

Reviewer #3: Yes

4. Is the manuscript presented in an intelligible fashion and written in standard English?

Reviewer #1: Yes

Reviewer #2: No

Reviewer #3: Yes

5. Review Comments to the Author

Reviewer #1: The authors have performed an interesting study on natural sound from bird species in an urban environment, using well-known and standard acoustic indices. They obtain robust results using pre-processing data by analysing cross-correlations between indices and bird species, thus discerning more accurately their effective applications to classify such natural sounds. While I can recommend the publication of the paper in PLOS ONE, I have few suggestions which the authors may consider for improvement of the presentation.

(1) Fig.3: The data for ADI and 1-AEI look identical. Is there a reason for that coincidence? Similar observation holds for H and Hf. Can you give us a hint for this?

(2) The different correlations observed at different sites should also reflect the different local noise conditions. Can you perhaps estimate such effects in order to understand the variations of correlation values found?

(3) There is quite recent literature about birds sounds in urban environments that the authors seem not to be aware of. I recommend to check the following references:

(3a) Benocci, et al. Mapping of the Acoustic Environment at an Urban Park...

Sensors 22, 3528 (2022).

(3b) Benocci, R., et al. Auto-correlations and long time memory of environment sound..

Ecological Indicators 134, 108492 (2022).

Reviewer #2: The study examines effectiveness of acoustic indices in bird species richness estimation in urban area. Similar studies have been conducted in various environments, so it is not novel approach. However inconsistent results reported in various studies point out that we still need more examples of such comparisons to understand where and when acoustic indices can be used as an approximation of biodiversity. From this perspective the study is valuable.

Authors identified bird species in 840 5-min sound samples (collected across more than one year) by manual spectrogram scanning and listening to recordings, calculated acoustic indices, and compared it by using Spearman’s correlation. This is very simple approach and, in my opinion, does not fully explore data. Authors reported weak or insignificant correlations between bird species richness and acoustic indices. The limitation of the study is sample size (only 3 recording points). The strong side of the study is that authors sampled soundscape across whole year. There is not a lot of studies analysing effectiveness of acoustic indices across a day and season. I think that daily and seasonally variable effectiveness of acoustic indices may be the main reason of found weak correlations. Did you check how well work acoustic indices when you compare separately dawn chorus, day and dusk chorus, or breeding vs non-breeding season? I suggest re-analysing data that way. I do not know how sound samples are distributed across day and season, maybe it will be necessary to analyse more sound samples to get optimal distribution, but authors can examine how each acoustic index estimate bird species richness dependently on time in a day and season (at least breeding vs non-breeding season). They can do this in simple way applying GLMMs in which they will specify the index for each sound sample as a dependent variable, number of detected bird species, time in a season, time in a day and one-way interactions as fixed effects, point id as a random effect. At the current stage of analysis, in my opinion, it will be difficult to publish results of the study. Also description of the methods should be more precise. I suggest considering reorganization of discussion and describing all indices together instead each of them separately. I suggest adding raw data as a supplementary materials.

More detailed comments:

L36: …840, 5-min sound samples.

L36-37: You used raw and pre-processed data.

L37: Spearman’s correlation

L38: …manual spectrogram scanning and listening to recordings.

L47: Please remove solely.

L52-53: Mobile network also serves such possibility and seems to be more useful than wi-fi.

L60-61: Not only these packages and not only R, but I agree that calculation is easy.

L64-65: Please remember that some indices correlate negatively with soundscape complexity.

L66-67: Please look at this review https://doi.org/10.1111/brv.12890, it can be useful.

L71-80: Consider mentioning about complexity and variation of birds vocalization – two different species (or even individuals within a species) will generate different values of acoustic indices.

L78-80: Great point. But also specific species composition may be one of the main factors responsible for effectiveness of acoustic indices.

L91: Not many papers reported strong correlation, it is rather moderate or weak.

L96: ...is home to 42 bird species, including endemic...

L96-98: How many of them are breeding species, how many are sedentary? Please write how long is a breeding season and when it has place.

L108-109: The distance between recorders was ca 100 m what means that you probably recorded the same individuals from different recording points (individuals between point A, B and C probably were recorded by three recorders). Most studies report detection distance for most of songbirds ranging between 100 and 200 m. Do you have information about detection distance of used by you equipment? Recording of the same individuals from various points do not limits your study, because soundscape recorded in each point is different.

L121-122: Please write about frequency response, signal to noise ratio and detection distance of recorders. Did you calibrate recorders before using?

L124:125: Impressive. Do you use these data in silent cities project?

L129-130: 840 five minutes sound samples. Please describe how did you select recording? Was it random choose? How was the distribution of sound samples across the year (because you recorded soundscape more than year, does it mean that months from July to June were sampled twice?)? Was it the same number of sound samples analysed per recording point? Did you validate recordings before analyzing (quality, noise, species richness)?

L130: …daily activity period, including dawn and dusk chorus.

L132: Which software and settings of spectrogram did you use? How many observers analysed recordings and if they were experienced in birds vocalisations? Did you recognise the species based on songs, calls or both? This is important to point out, because in the breeding season the same species richness will generate different complexity of soundscape than in non-breeding season.

L136-137: How commonly vocalizations of animals other than birds were recorded? The information about other biological sources of sound in your study area is missing.

L140: Similar effect you can get modifying setting of acoustic indices calculation.

L150-154: Please add a table (e.g., as a supply mat) in which you will show settings for each index.

L155-159: This point is not clear. Did you calculate indices for 5-min sound samples? If yes, you had constant sound-sample duration and you can use raw or standarised indices values for further analyses. Please be precise, describing the dataset used for manual spectrogram scanning and listening to recordings (840 sound samples) and calculating indices (840 sound samples or whole dataset).

L138: Please look at this review: https://doi.org/10.1080/09524622.2021.2010598

L175-176: So these two indices should correlate negatively and positively with bird species richness.

L197-199: It is very simple method. I think that you could use more sophisticated analyses to show how acoustic indices estimate bird species richness dependently on time in a day and time in a season. For example you can use GLMM for each index to check how the index predict bird species richness dependently on time in a day, season and recording point. Such analysis will considerably improve sounds of your study. When you use Spearman’s rho you should demonstrate evidence of a monotonic relationship and lack of severe skewness in raw data. If I good understood you compared the number of bird species detected manually in 840 sound samples with many acoustic indices calculated on the same 840 sound samples. I do not see test comparing effectiveness of indices on raw and pre-processing data, similarly like between recording points, but you write about it in discussion.

L202-203: 840 5-min sound samples? I suggest adding histogram (or modifying Table 1) to show how many sound samples each bird species was detected.

L205-206: Why?

L206-209: You do not need mention them here, please add marker in Table 1 which species are endemic.

L213: I suggest to show in how many sound samples each species was detected in each recording point instead markers.

L241: You showed the same in the table and on fig. It would be better to see raw data on the fig., e.g., number of bird species vs index value for each sound sample and the mean/median +/- 95%cu. If you would like to check difference in bird species richness estimation between recording points I suggest using GLMM with point ID as an effect. Additionally in such model you probably would be able to test the effect of time in a day and season on effectiveness of acoustic indices. Please consider such approach.

L247: In my opinion discussion needs reorganization. I suggest writing it more holistically (like in conclusions) instead describing each index separately. When you write whether pre-processing data increase effectiveness of the index in bird species richness estimation you need to support it by statistical test. More references to effectiveness of acoustic indices in estimation of bird species richness in urban areas are needed.

L249-255: In your study you did not test the effectiveness of soundscape recorders in comparison to human observers. Therefore this part of discussion is not necessary. You estimated bird species richness based on soundscape recording and compared it with acoustic indices, what is appropriate approach.

L343-344: There was a few studies examining effectiveness of acoustic indices in urban areas from other regions. Please look at them.

Reviewer #3: The paper “Acoustic indices as proxies for bird species richness in an urban green space in Metro Manila” describes the correlation results between bird species richness and eco-acoustic indices.

The paper is clear but requires supplementary analysis before being accepted. It focuses just on a specific aspect of the soundscape recorded at three sites linked to bird species richness. Generally, eco acoustic indices represent summaries of the whole soundscape and highlight specific features of the frequency spectrum. This concept should be included in the introduction.

More in detail in the Material and Methods section (Data collection), you should include the sampling rate of the recorder and the frequency response. The latter is important especially for low-cost sensors.

Sect. Manual aural identification: Here, report also all other sound sources identified and their frequencies. This is important especially in urban green areas

Sect. Acoustic index analysis: when you introduce the pre-processing, please report an example of a spectrum for a raw and pre-processed audio file.

All indices are based upon a FFT computation of audio recordings. Please specify the number of FFT points use, the frequency and time resolution.

In Sect ADI and IAEI , it is reported that each frequency bin represents a particular bird species. This is NOT true!!

At line 176, I would say that higher ADI and 1-AEI indicate higher frequency occupation of the spectrum.

Fig 2 is not clear: As ordinate axes use a clearer title. Here you should report also differences among the sites in terms of richness and abundance of bird species. This is important because the correlation between indices depends also on the species abundance.

If possible include a temporal trend of the mean bird richness as well as for the eco-acoustic indices.

Lines 264-266: this part needs to be improved

Line 272-273. This is why abundance of birds should be included in the analysis.

Line 303-304: this is why you need to include a more detailed aural survey.

In the Conclusions, you should also stress that, besides CNN and machine learning, an approach which implies the use of statistics is also able to help discriminate among different soundscapes in urban areas based upon eco-acoustic indices, as reported in:

Benocci, R.; Roman, H.E.; Bisceglie, A.; Angelini, F.; Brambilla, G.; Zambon, G. Eco-Acoustic Assessment of an Urban Park by Statistical Analysis. Sustainability 2021, 13, 7857.

Benocci, R.; Brambilla, G.; Bisceglie, A.; Zambon, G. Eco-Acoustic Indices to Evaluate Soundscape Degradation Due to Human Intrusion. Sustainability 2020, 12, 10455.

6. PLOS authors have the option to publish the peer review history of their article (what does this mean?). If published, this will include your full peer review and any attached files.

Reviewer #1: No

Reviewer #2: No

Reviewer #3: No

---

## [Author Response · Author response to Decision Letter 0]

2 Apr 2023

Author Rebuttals to Initial Comments:

Reviewer #1

The authors have performed an interesting study on natural sound from bird species in an urban environment, using well-known and standard acoustic indices. They obtain robust results using pre-processing data by analysing cross-correlations between indices and bird species, thus discerning more accurately their effective applications to classify such natural sounds. While I can recommend the publication of the paper in PLOS ONE, I have few suggestions which the authors may consider for improvement of the presentation.

(1) Fig.3: The data for ADI and 1-AEI look identical. Is there a reason for that coincidence? Similar observation holds for H and Hf. Can you give us a hint for this?

The ADI and AEI divide the spectrogram into frequency bins and take the proportion of amplitude values in each bin above a dB threshold [15]. The ADI applies the Shannon diversity index to these values, whereas the AEI applies the Gini index. A higher Shannon’s index indicates higher diversity, whereas a higher Gini index indicates lower evenness and diversity. ADI values should correlate positively and AEI values negatively with bird species richness.

We calculated the inverse of AEI, herein referred to as the inverse Acoustic Evenness Index (1-AEI) to facilitate comparability between the two indices [21]. Higher ADI and 1-AEI indicate higher frequency occupation of the spectrum and higher sound diversity. Thus, Spearman’s correlations for ADI and 1-AEI look identical. 

The H is calculated as the product of Ht and Hf, where Ht represents the Shannon entropy of the amplitude envelope, and Hf represents the Shannon entropy of the frequency spectrum [17]. The Spearman’s correlations for H and Hf look identical, as they are both based on spectral complexity, whereas the Ht is based on amplitude parameters.

(2) The different correlations observed at different sites should also reflect the different local noise conditions. Can you perhaps estimate such effects in order to understand the variations of correlation values found?

To estimate the effects of noise on the acoustic indices, we generated 5-min Brownian noise in increasing 0.01 dB increments in Audacity and mixed each clip with 5 randomly selected raw 5-min sound samples from our dataset from each site from 8-9 am. This resulted in 70 mixed tracks per site. We calculated the acoustic indices for each mixed track to see if noise affected their performance. We found that the nACI, H, Ht, and Hf were relatively robust to noise. 

(3) There is quite recent literature about birds sounds in urban environments that the authors seem not to be aware of. I recommend to check the following references:

(3a) Benocci, et al. Mapping of the Acoustic Environment at an Urban Park...

Sensors 22, 3528 (2022).

Similar to the current study, Benocci et al. (2022) used low-cost recorders. They evaluated the frequency response of each recorder in terms of its sensitivity at different frequencies. They did this by exposing the recorders to white noise generated by a loudspeaker from a fixed distance.

We characterized our recorder’s frequency response and included this in Line 153-160 of the revised manuscript and included the details in S1 Fig. We found that the frequency response was not flat over the entire frequency range, and higher frequencies were increasingly attenuated, except for recorder A. 

(3b) Benocci, R., et al. Auto-correlations and long time memory of environment sound..

Ecological Indicators 134, 108492 (2022).

Benocci et al. (2022) used GLMMs to evaluate the relationship between acoustic indices and rainfall frequency and intensity. 

We used GLMMs to determine how acoustic indices predict bird species richness depending on the season and time of day. We included this in Line 265-277 of the materials and methods section of the revised manuscript. The GLMM results can be found in Table 4 and S2 Table. The detailed results can be found in Line 344-351 of the revised manuscript.

We found that 1-AEI, Ht, and AR predicted bird species richness in pre-processed 5-min sound samples, depending on the season and time of day. The nACI also predicted bird species richness but independently of season and time of day. Conversely, season and time of day were stronger predictors for ADI, BI, H, and Hf.

Reviewer #2

The study examines effectiveness of acoustic indices in bird species richness estimation in urban area. Similar studies have been conducted in various environments, so it is not novel approach. However inconsistent results reported in various studies point out that we still need more examples of such comparisons to understand where and when acoustic indices can be used as an approximation of biodiversity. From this perspective the study is valuable.

Authors identified bird species in 840 5-min sound samples (collected across more than one year) by manual spectrogram scanning and listening to recordings, calculated acoustic indices, and compared it by using Spearman’s correlation. This is very simple approach and, in my opinion, does not fully explore data. Authors reported weak or insignificant correlations between bird species richness and acoustic indices. The limitation of the study is sample size (only 3 recording points). The strong side of the study is that authors sampled soundscape across whole year. There is not a lot of studies analysing effectiveness of acoustic indices across a day and season. I think that daily and seasonally variable effectiveness of acoustic indices may be the main reason of found weak correlations. 

Did you check how well work acoustic indices when you compare separately dawn chorus, day and dusk chorus, or breeding vs non-breeding season? I suggest re-analysing data that way. I do not know how sound samples are distributed across day and season, maybe it will be necessary to analyse more sound samples to get optimal distribution, but authors can examine how each acoustic index estimate bird species richness dependently on time in a day and season (at least breeding vs non-breeding season). They can do this in simple way applying GLMMs in which they will specify the index for each sound sample as a dependent variable, number of detected bird species, time in a season, time in a day and one-way interactions as fixed effects, point id as a random effect. At the current stage of analysis, in my opinion, it will be difficult to publish results of the study. Also description of the methods should be more precise. I suggest considering reorganization of discussion and describing all indices together instead each of them separately. 

I suggest adding raw data as a supplementary materials.

Yes, we agree. We included the results of the acoustic index analysis for 840 raw and 840 pre-processed 5-min sound samples as supplementary materials (S3 and S4 Tables) and in Line 318-325 of the revised text.

More detailed comments:

We selected 840 5-min sound samples, equivalent to 70 hours, through stratified sampling and pre-processed them before conducting acoustic index analysis on the raw and pre-processed data. We measured Spearman’s correlation between each acoustic index and bird species richness obtained from manual spectrogram scanning and listening to recordings.

L36: …840, 5-min sound samples.

L36-37: You used raw and pre-processed data.

L37: Spearman’s correlation

L38: …manual spectrogram scanning and listening to recordings.

We included the suggestions and reworded our manuscript to read as:

“We selected 840 5-min sound samples, equivalent to 70 hours, through stratified sampling and pre-processed them before conducting acoustic index analysis on the raw and pre-processed data. We measured Spearman’s correlation between each acoustic index and bird species richness obtained from manual spectrogram scanning and listening to recordings.”

L47: Please remove solely.

 Yes, we have removed “solely.” in this line.

L52-53: Mobile network also serves such possibility and seems to be more useful than wi-fi.

Thank you for the suggestion. We have added “mobile network.”

L60-61: Not only these packages and not only R, but I agree that calculation is easy.

We used the soundecology and seewave R packages, as they were the most used in studies to calculate acoustic indices although there are other ways to calculate these as the reviewer pointed out.

L64-65: Please remember that some indices correlate negatively with soundscape complexity.

Yes, we agree with the reviewer. We want to clarify that the reference studies for acoustic indices in our study reported positive correlations.

L66-67: Please look at this review https://doi.org/10.1111/brv.12890, it can be useful.

We thank the reviewer for this suggestion, and we added this reference as suggested and considered this in our rewording of our texts.

L71-80: Consider mentioning about complexity and variation of birds vocalization – two different species (or even individuals within a species) will generate different values of acoustic indices.

This is true. However, we are doing a community-level analysis. We take into account the call or song variation among individuals within a species by taking several samples over different times and different 5-min samples. 

L78-80: Great point. But also specific species composition may be one of the main factors responsible for effectiveness of acoustic indices.

The birds in the NSC are mostly passerines, so their frequency ranges have a high chance of overlap due to their relatively similar body sizes and vocal capacities.

L91: Not many papers reported strong correlation, it is rather moderate or weak.

Yes, we agree. We reworded our hypothesis to this: 

“We hypothesized that if acoustic index values are significantly correlated with the number of species heard, then the selected indices can be used as proxies for bird species richness in the field.”

L96: ...is home to 42 bird species, including endemic...

 We have corrected this.

L96-98: How many of them are breeding species, how many are sedentary? Please write how long is a breeding season and when it has place.

In our results, we reported that out of 21 species heard in the NSC, seven are Philippine endemics, thirteen are resident species, and only one is a migrant, the Brown Shrike. 

The breeding season for most birds in the Philippines aligns with the dry season, from November to April. The wet season is from May to October. We used the dry and wet seasons as a proxy for breeding and non-breeding season. We used this as a factor in our GLMMs. 

L108-109: The distance between recorders was ca 100 m what means that you probably recorded the same individuals from different recording points (individuals between point A, B and C probably were recorded by three recorders). Most studies report detection distance for most of songbirds ranging between 100 and 200 m. Do you have information about detection distance of used by you equipment? Recording of the same individuals from various points do not limits your study, because soundscape recorded in each point is different.

L121-122: Please write about frequency response, signal to noise ratio and detection distance of recorders. Did you calibrate recorders before using?

We tested the response of each low-cost recorder across a range of frequencies. Using Audacity, we generated a 30-sec tone from 1 Hz to 10,000 Hz and played it through a loudspeaker held at breast level, gradually increasing the distance from the recorder. We repeated this several times and we plotted the spectrum data of the 30-sec generated tone against the average response of each recorder (S1 Fig). The frequency response was not flat over the entire frequency range, except for recorder A, and higher frequencies were increasingly attenuated (S1 Fig).

L124:125: Impressive. Do you use these data in silent cities project?

The data was not used in the Silent Cities Project. However, initially, we set-up recorders in the National Science Complex to study the effects of the anthropause on the bird community. But, we still have insufficient pre-pandemic and post-pandemic data. The three recorders were continuously recording until July 2022. The Site A system resumed recording in November 2022 and is still recording now.

L129-130: 840 five minutes sound samples. Please describe how did you select recording? Was it random choose? How was the distribution of sound samples across the year (because you recorded soundscape more than year, does it mean that months from July to June were sampled twice?)? Was it the same number of sound samples analysed per recording point? Did you validate recordings before analyzing (quality, noise, species richness)?

L130: …daily activity period, including dawn and dusk chorus.

We randomly selected 840 .wav files from the database based on the YYYY-MM-DD-HH-mm timestamp. Through stratified sampling, we selected 20 sound samples from each hour between 5:00 AM and 6:00 PM to cover the daily activity period, including the dawn and dusk chorus. This resulted in a total of 280 samples for each recording point. 

The distribution of the sound samples across months is variable but fully represented (see Figure in attached Response to Reviewers). Yes, the months of July to January were sampled twice.

L132: Which software and settings of spectrogram did you use? How many observers analysed recordings and if they were experienced in birds vocalisations? Did you recognise the species based on songs, calls or both? This is important to point out, because in the breeding season the same species richness will generate different complexity of soundscape than in non-breeding season.

We viewed each pre-processed 5-min sound sample in Audacity, using Spectrogram View set to Linear Scale, with the Minimum Frequency set to 0 kHz and the Max Frequency set to 8 kHz (Fig 2). S. Diaz was the primary observer who analyzed all the recordings to reduce variability between observers. 

We recognized species based on their calls and songs since the seasonal variation is less pronounced in the Philippines than in temperate countries. The Philippines has two seasons: dry and wet. We used the dry and wet seasons as a proxy for breeding and non-breeding season.

L136-137: How commonly vocalizations of animals other than birds were recorded? The information about other biological sources of sound in your study area is missing.

In addition to bird sounds, we also detected various other biotic sounds, including those produced by mammals, amphibians, reptiles, and insects. These include the sounds of chickens, dogs, geckos, frogs, sheep, and crickets. We also identified anthrophonies, such as human speech, construction noise, and road traffic, as well as geophonies, such as rain and wind. Although we detected other non-avian sounds in our recordings, we did not take their frequencies into account.

L138: Please look at this review: https://doi.org/10.1080/09524622.2021.2010598

We considered this in rewriting the manuscript and included it in the references.

L140: Similar effect you can get modifying setting of acoustic indices calculation.

We wanted to use the default settings of the acoustic indices to streamline the processing of multiple .wav files.

L150-154: Please add a table (e.g., as a supply mat) in which you will show settings for each index.

We have added the summary of parameters used in the calculation of acoustic indices as supplementary materials (S1 Table).

L155-159: This point is not clear. Did you calculate indices for 5-min sound samples? If yes, you had constant sound-sample duration and you can use raw or standarised indices values for further analyses. Please be precise, describing the dataset used for manual spectrogram scanning and listening to recordings (840 sound samples) and calculating indices (840 sound samples or whole dataset).

We used the 840 pre-processed sound samples for manual spectrogram scanning and listening to recordings. 

We used two data sets for the acoustic index analysis: 840 raw and 840 pre-processed .wav files. The only difference between the two sets of 840 5-min sound samples is that the pre-processed samples were subjected to processing steps (i.e., high-pass filter, noise reduction, amplification). Both datasets have the same YYYY-MM-DD-HH-mm timestamp and recording point. 

These were written into the revised manuscript in Line 174-175 and Line 196-197.

L175-176: So these two indices should correlate negatively and positively with bird species richness.

Yes, ADI values should correlate positively and AEI values negatively with bird species richness. Thus, to facilitate comparability between the two indices, we decided to get the inverse of AEI, herein referred to as the 1-AEI.

L197-199: It is very simple method. I think that you could use more sophisticated analyses to show how acoustic indices estimate bird species richness dependently on time in a day and time in a season. For example you can use GLMM for each index to check how the index predict bird species richness dependently on time in a day, season and recording point. Such analysis will considerably improve sounds of your study. When you use Spearman’s rho you should demonstrate evidence of a monotonic relationship and lack of severe skewness in raw data. If I good understood you compared the number of bird species detected manually in 840 sound samples with many acoustic indices calculated on the same 840 sound samples. I do not see test comparing effectiveness of indices on raw and pre-processing data, similarly like between recording points, but you write about it in discussion.

L241: You showed the same in the table and on fig. It would be better to see raw data on the fig., e.g., number of bird species vs index value for each sound sample and the mean/median +/- 95%cu. If you would like to check difference in bird species richness estimation between recording points I suggest using GLMM with point ID as an effect. Additionally in such model you probably would be able to test the effect of time in a day and season on effectiveness of acoustic indices. Please consider such approach.

We thank the reviewer for this suggestion and have used generalized linear mixed models (GLMMs) to examine how acoustic indices predict bird species richness in pre-processed 5-min sound samples depending on season and time of day. For each sound sample, we specified the acoustic index as the dependent variable, bird species richness, season, and time of day as fixed effects, and the recording point as the random effect. We fitted a set of three models for each acoustic index, which included the full model and fixed effects models in R using the lme4 and glmmTMB packages (S2 Table).

L202-203: 840 5-min sound samples? I suggest adding histogram (or modifying Table 1) to show how many sound samples each bird species was detected.

L206-209: You do not need mention them here, please add marker in Table 1 which species are endemic.

L213: I suggest to show in how many sound samples each species was detected in each recording point instead markers.

We thank the reviewer for this suggestion and have modified Table 1 to reflect how many sound samples each bird species was detected in each recording point. In Table 1, we marked Philippine endemic species with asterisk (*).

L205-206: Why?

 There is insufficient eBird data for both species. 

L247: In my opinion discussion needs reorganization. I suggest writing it more holistically (like in conclusions) instead describing each index separately. When you write whether pre-processing data increase effectiveness of the index in bird species richness estimation you need to support it by statistical test. More references to effectiveness of acoustic indices in estimation of bird species richness in urban areas are needed.

Thank you for this suggestion. We reorganized the discussion and reported the acoustic index correlations more holistically. We calculated Fisher's z-transformation to compare the correlations between the raw and pre-processed datasets.

L249-255: In your study you did not test the effectiveness of soundscape recorders in comparison to human observers. Therefore this part of discussion is not necessary. You estimated bird species richness based on soundscape recording and compared it with acoustic indices, what is appropriate approach.

 Yes, we agree. We removed this part of the discussion. 

L343-344: There was a few studies examining effectiveness of acoustic indices in urban areas from other regions. Please look at them.

We wanted to mention that this is the first study testing acoustic indices as proxies for bird species richness in an urban setting in the Philippines. We are aware that other studies used acoustic indices in urban areas in temperate regions.

Reviewer #3:

The paper “Acoustic indices as proxies for bird species richness in an urban green space in Metro Manila” describes the correlation results between bird species richness and eco-acoustic indices.

The paper is clear but requires supplementary analysis before being accepted. It focuses just on a specific aspect of the soundscape recorded at three sites linked to bird species richness. 

Generally, eco acoustic indices represent summaries of the whole soundscape and highlight specific features of the frequency spectrum. This concept should be included in the introduction.

Thank you for the suggestion. We’ve added this to the introduction: “Several ecoacoustic indices were proposed to summarize the acoustic features of soundscapes using the sound spectrum [8]. Moreover, acoustic indices are mathematical functions that measure the distribution of acoustic energy across time and frequency in a recording [9]. These indices enable the quick processing of sound data and measure the acoustic complexity, diversity, evenness, entropy, and richness of recordings. They are based on species diversity indices and are thought to characterize the variation in sound production in animal communities across time, serving as proxies for diversity metrics such as species abundance, richness, evenness, and diversity [10].”

More in detail in the Material and Methods section (Data collection), you should include the sampling rate of the recorder and the frequency response. The latter is important especially for low-cost sensors.

Again, we thank the reviewer for this suggestion. In the Materials and methods section, we noted that we used a sampling rate of 44.1 kHz, and we also described the frequency response of the recorders.

Sect. Manual aural identification: Here, report also all other sound sources identified and their frequencies. This is important especially in urban green areas

In addition to bird sounds, we also detected various other biotic sounds, including those produced by mammals, amphibians, reptiles, and insects. These include the sounds of chickens, dogs, geckos, frogs, sheep, and crickets. We also identified anthrophonies, such as human speech, construction noise, and road traffic, as well as geophonies, such as rain and wind. Although we detected other non-avian sounds in our recordings, we did not take their frequencies into account.

Sect. Acoustic index analysis: when you introduce the pre-processing, please report an example of a spectrum for a raw and pre-processed audio file.

Yes, we agree. We added sample spectrograms for raw and pre-processed .wav files opened in Audacity (Fig 2).

All indices are based upon a FFT computation of audio recordings. Please specify the number of FFT points use, the frequency and time resolution.

We specified in the Materials and methods section that we used a sampling frequency of 44,100 Hz and a fast Fourier transform (FFT) window of 512 points (S1 Table). This corresponds to a frequency resolution of FR = 86.133 Hz and a time resolution of TR = 0.0116 s.

In Sect ADI and IAEI , it is reported that each frequency bin represents a particular bird species. This is NOT true!!

At line 176, I would say that higher ADI and 1-AEI indicate higher frequency occupation of the spectrum.

We thank the reviewer for finding this error, and we have corrected it. We changed it to what we originally meant: “Higher ADI and 1-AEI should indicate higher frequency occupation of the spectrum and higher sound diversity.”

Fig 2 is not clear: As ordinate axes use a clearer title. Here you should report also differences among the sites in terms of richness and abundance of bird species. This is important because the correlation between indices depends also on the species abundance.

If possible include a temporal trend of the mean bird richness as well as for the eco-acoustic indices.

We thank the reviewer for this suggestion. We modified the figure (See Figure 3) to reflect the differences among the sites in terms of richness, and we used a clearer title. We also included a temporal trend of the mean bird richness (See Figure 4).

Lines 264-266: this part needs to be improved

Yes, we agree. We changed it to this: “Ground-truthing of the acoustic indices should be conducted before applying them to studies in different environments as their performance depends on various factors including, the local noise conditions, bird species composition of the site, season, and time of day.”

Line 272-273. This is why abundance of birds should be included in the analysis.

The limitation of our study is that we did not measure bird species abundance because of the difficulty of telling apart individuals based simply on their calls and the pandemic limited on-site observations simultaneous with the recordings. Hence, the study focused on the relationship of acoustic indices with bird species richness.

Given the proximity of the three sites, the bird community across them is most likely shared. We assume that the bird abundance and bird richness are similar across the three sites.

Line 303-304: this is why you need to include a more detailed aural survey.

We mentioned in the results that in addition to bird sounds, we also detected various other biotic sounds, including those produced by mammals, amphibians, reptiles, and insects. Although we detected other non-avian sounds in our recordings, we did not take their frequencies into account, as we wanted to focus on bird vocalizations.

In the Conclusions, you should also stress that, besides CNN and machine learning, an approach which implies the use of statistics is also able to help discriminate among different soundscapes in urban areas based upon eco-acoustic indices, as reported in:

Benocci, R.; Roman, H.E.; Bisceglie, A.; Angelini, F.; Brambilla, G.; Zambon, G. Eco-Acoustic Assessment of an Urban Park by Statistical Analysis. Sustainability 2021, 13, 7857.

Benocci, R.; Brambilla, G.; Bisceglie, A.; Zambon, G. Eco-Acoustic Indices to Evaluate Soundscape Degradation Due to Human Intrusion. Sustainability 2020, 12, 10455.

We added these references as suggested and note their importance. In the Conclusions, we added, “The use of other analysis, such as principal component analysis (PCA) [55] and cluster analysis [56], also help discriminate among different soundscapes in urban areas based on ecoacoustic indices.”

---

## [Decision Letter · Decision Letter 1]

26 Jun 2023

PONE-D-22-34645R1Acoustic indices as proxies for bird species richness in an urban green space in Metro ManilaPLOS ONE

Dear Dr. Diaz,

Thank you for submitting your manuscript to PLOS ONE. After careful consideration, we feel that it has merit but does not fully meet PLOS ONE’s publication criteria as it currently stands. Therefore, we invite you to submit a revised version of the manuscript that addresses the points raised during the review process.

Please remove gradient from the caption and specify min and max values in dB, in Fig. 2. Also, please address the comment from reviewer 3, with respect to differences among the used sensors.

We look forward to receiving your revised manuscript.

Kind regards,

Luca Nelli, PhD

Academic Editor

PLOS ONE

Journal Requirements:

Additional Editor Comments:

Please remove gradient from the caption and specify min and max values in dB, in Fig. 2.

Also, please address the comment from reviewer 3, with respect to differences among the used sensors.

Reviewers' comments:

Reviewer's Responses to Questions

**Comments to the Author**

1. If the authors have adequately addressed your comments raised in a previous round of review and you feel that this manuscript is now acceptable for publication, you may indicate that here to bypass the “Comments to the Author” section, enter your conflict of interest statement in the “Confidential to Editor” section, and submit your "Accept" recommendation.

Reviewer #1: All comments have been addressed

Reviewer #3: (No Response)

2. Is the manuscript technically sound, and do the data support the conclusions?

Reviewer #1: Yes

Reviewer #3: Yes

3. Has the statistical analysis been performed appropriately and rigorously? 

Reviewer #1: Yes

Reviewer #3: Yes

4. Have the authors made all data underlying the findings in their manuscript fully available?

Reviewer #1: Yes

Reviewer #3: Yes

5. Is the manuscript presented in an intelligible fashion and written in standard English?

Reviewer #1: Yes

Reviewer #3: Yes

6. Review Comments to the Author

Reviewer #1: The authors have satisfactorily responded to all pints raised. As such, I can recommend the publication of this paper in its present form.

Reviewer #3: The authors mostly addressed the reviewer’s suggestions. Still there are some other minor points to be clarified and stressed.

Fig 2: remove gradient from the caption and specify min and max values in dB.

The frequency response that you obtained reveals that there could be significant differences among the used sensors. This may imply that also the indices calculation could be affected by it. This limitation needs to appear in the Conclusions.

7. PLOS authors have the option to publish the peer review history of their article (what does this mean?). If published, this will include your full peer review and any attached files.

Reviewer #1: No

Reviewer #3: **Yes: **Giovanni Zambon

---

## [Author Response · Author response to Decision Letter 1]

6 Jul 2023

Reviewer #1

The authors have satisfactorily responded to all points raised. As such, I can recommend the publication of this paper in its present form.

We thank the reviewer for their comments and suggestions, which greatly improved this manuscript.

Reviewer #3

The authors mostly addressed the reviewer’s suggestions. Still there are some other minor points to be clarified and stressed.

Fig 2: remove gradient from the caption and specify min and max values in dB.

Thank you for the suggestion. We have removed the gradient from the caption and specified the minimum and maximum values in dB. The amplitude ranges from -60 dB to -44 dB (raw) and -60 dB to -29 dB (pre-processed).

The frequency response that you obtained reveals that there could be significant differences among the used sensors. This may imply that also the indices calculation could be affected by it. This limitation needs to appear in the Conclusions.

We have included this limitation of low-cost recorders in the Conclusions and it reads as follows “As the frequency response of our recorders was not flat over the entire frequency range, which impacts the calculation of acoustic indices, low-cost recorders should be calibrated before use in ecoacoustic studies to offset this limitation. We also recognize the importance of pre-processing sound data, such as applying high-pass filters, noise reduction, and signal amplification, to account for differences among recorders, remove unwanted ambient sounds, and improve the detection of bird species.”

---

## [Editor Report · Decision Letter 2]

10 Jul 2023

Acoustic indices as proxies for bird species richness in an urban green space in Metro Manila

PONE-D-22-34645R2

Dear Dr. Diaz,

We’re pleased to inform you that your manuscript has been judged scientifically suitable for publication and will be formally accepted for publication once it meets all outstanding technical requirements.

Kind regards,

Luca Nelli, PhD

Academic Editor

PLOS ONE
---

## [Editor Report · Acceptance letter]

20 Jul 2023

PONE-D-22-34645R2 

Acoustic indices as proxies for bird species richness in an urban green space in Metro Manila 

Dear Dr. Diaz:

I'm pleased to inform you that your manuscript has been deemed suitable for publication in PLOS ONE. Congratulations! Your manuscript is now with our production department. 

Kind regards, 

on behalf of

Dr. Luca Nelli 

Academic Editor

PLOS ONE